# Tamoxifen resistance alters sensitivity to 5-fluorouracil in a subset of estrogen receptor-positive breast cancer

Takayuki Watanabe[1], Takaaki Oba[1], Keiji Tanimoto[2], Tomohiro Shibata[1], Shinobu Kamijo[1], Ken-ichi Ito[1]*

1 Division of Breast and Endocrine Surgery, Department of Surgery, Shinshu University School of Medicine, Matsumoto, Nagano, Japan, 2 Department of Radiation Medicine, Research Institute for Radiation Biology and Medicine, Hiroshima University, Hiroshima City, Hiroshima, Japan

* kenito@shinshu-u.ac.jp

**Data Availability Statement:** All relevant data are within the manuscript and its Supporting Information files.

## Abstract

Sequential treatment with endocrine or chemotherapy is generally used in the treatment of estrogen receptor (ER)-positive recurrent breast cancer. To date, few studies have investigated the effect of long-term endocrine therapy on the response to subsequent chemotherapy in ER-positive breast cancer. We examined whether a preceding endocrine therapy affects the sensitivity to subsequent chemotherapy in ER-positive breast cancer cells. Three ER-positive breast cancer cell lines (T47D, MCF7, BT474) and tamoxifen-resistant sublines (T47D/T, MCF7/T, BT474/T) were analyzed for sensitivity to 5-fluorouracil, paclitaxel, and doxorubicin. The mRNA levels of factors related to drug sensitivity were analyzed by RT-PCR. MCF7/T cells became more sensitive to 5-fluorouracil than wild-type (wt)-MCF7 cells. In addition, the apoptosis induced by 5-fluorouracil was significantly increased in MCF7/T cells. However, no difference in sensitivity to chemotherapeutic agents was observed in T47D/T and BT474/T cells compared with their wt cells. Dihydropyrimidine dehydrogenase (*DPYD*) mRNA expression was significantly decreased in MCF7/T cells compared with wt-MCF7 cells. The expression of *DPYD* mRNA was restored with 5-azacytidine treatment in MCF7/T cells. In addition, *DPYD* 3′-UTR luciferase activity was significantly reduced in MCF7/T cells. These data indicated that the expression of *DPYD* mRNA was repressed by methylation of the *DPYD* promoter region and post-transcriptional regulation by miRNA in MCF7/T cells. In the mouse xenograft model, capecitabine significantly reduced the tumor volume in MCF7/T compared with MCF7. The results of this study indicate that endocrine therapy could alter the sensitivity to chemotherapeutic agents in a subset of breast cancers, and 5-fluorouracil may be effective in tamoxifen-resistant breast cancers.

## Introduction

Breast cancer is the most common cancer type in women worldwide, accounting for approximately one quarter of all female cancers [1, 2]. Furthermore, breast cancer is the most common cause of cancer death in women, accounting for approximately 15% of cancer deaths [1, 2].

**Funding:** The authors received no specific funding for this work.

**Competing interests:** The authors have declared that no competing interests exist.

Estrogen receptor (ER)-positive breast cancers are the most frequently occuring type of breast cancer, accounting for around three quarters of all breast cancer cases in the world, and this percentage is even higher among older women [2, 3].

Endocrine therapy has been the mainstay therapy for ER-positive breast cancer, and it has been widely used as an adjuvant therapy. In addition to conventional endocrine therapy and cytotoxic chemotherapy, molecular-targeted agents, such as mechanistic target of rapamycin (mTOR) and cyclin-dependent kinase (CDK) 4/6 inhibitors, have been introduced for the treatment of metastatic or recurrent ER-positive breast cancers [4–8]. Furthermore, a poly ADP-ribose polymerase (PARP) inhibitor has been approved for *BRCA* mutation-positive patients with metastatic breast cancer [9]. Thus, treatments for recurrent ER-positive breast cancer have recently diversified rapidly.

For patients with ER-positive breast cancer who are at risk of recurrence, chemotherapy consisting of anthracycline and taxane followed by endocrine therapy has been conducted as standard adjuvant therapy [10]; however, when a patient experiences cancer relapse, the treatment strategy has to be decided on an individual basis by gathering and assessing clinical information, such as the response to previous endocrine therapies, status of metastatic organs, and interval until recurrence after completion of adjuvant endocrine therapy, as well as the patient's preferences, due to the lack of established biomarkers that reflect the biology of recurrent cancer in real time. Since the advent of CDK4/6 inhibitors, chemotherapy has become less likely to be used in early lines of treatment for recurrent ER-positive breast cancer, including cases that have recurred during adjuvant endocrine therapy with tamoxifen (TAM) or aromatase inhibitors. However, chemotherapy should be administered to patients with life-threatening metastases or *de novo* endocrine-resistant tumors.

Recently, a phase-3 trial was conducted comparing the effects of taxane and S-1 in the first-line treatment of patients with ER-positive and human epidermal growth factor receptor type 2 (HER2)-negative metastatic breast cancer. This cohort consisted of patients who had become resistant to endocrine therapy and had received no chemotherapy for advanced disease. In this trial, the efficacy of S-1 was shown to be at least as good as that of taxane with respect to overall survival [11]. Although no difference was observed between the efficacy of S-1 and taxane in this clinical trial, we expected that there should have been tumors that were more sensitive to either of them, owing to the diverse characteristics of endocrine-resistant recurrent breast cancer. Moreover, it may be possible that the biology of cancer cells that acquired resistance to endocrine therapeutic agents is different from that of primary cancer cells, and the change in the biology of cancer cells may alter the sensitivity to subsequently administered anticancer agents. However, few studies have investigated the effect of long-term endocrine therapy on the response to subsequent chemotherapy in ER-positive breast cancer to date, and there are no useful biomarkers for selecting drugs that could be more effective for each recurrent breast cancer.

The objective of this study was to examine whether preceding endocrine therapy could alter the sensitivity of ER-positive breast cancers to chemotherapeutic agents and explore biomarkers useful for personalized treatment of endocrine therapy-resistant recurrent cancer. We established TAM-resistant sublines in three ER-positive breast cancer cell lines and analyzed their sensitivities to chemotherapeutic agents.

## Materials and methods

### Cell lines and agents

ER-positive breast cancer cell lines (T47D, MCF7, BT474) were purchased from the American Type Cell Collection (Manassas, VA, USA) at the beginning of the study. All cell lines were

cultured in RPMI (Sigma-Aldrich, St. Louis, MO, USA) with 10% fetal bovine serum (FBS) at 37.0˚C and 5% $CO_2$. TAM-resistant sublines were established by continuous exposure to step-wise increases in the concentrations of TAM for more than 6 months and by using the limiting dilution method, during which time the medium was replaced every 3 d, and the cultured cells were subcultured after trypsinization when the cells reached 70% confluence. Through this process, we selected several TAM-resistant clones for each breast cancer cell line and then used one representative clone in subsequent experiments. TAM-resistant cell lines were designated as MCF7/T, T47D/T, and BT474/T. TAM, paclitaxel, doxorubicin, and 5-fluorouracil were purchased from Sigma-Aldrich.

## Cell proliferation assay

The cells were grown in six-well plates, and the number of viable cells following drug treatment was counted using CYTORECON (GE Healthcare Life Science, Tokyo, Japan). Cells ($1 \times 10^5$ cells/well) were seeded in six-well plates and incubated for 24 h. Then, 2 mL of medium with 1–30 μM of TAM was added into each well. After each indicated period, the cell numbers were directly counted.

## Apoptosis analysis

Cells were plated in six-well plates at a density of $5 \times 10^4$ cells/well. After 24 h, cells were treated with anticancer drugs and were cultured for another 48 h. To detect apoptotic cell death, DNA fragmentation was detected using Cell Death Detection ELISA[plus] (Roche Applied Science, Tokyo, Japan) following the manufacturer's instructions.

## WST assay

The growth inhibitory effects of tamoxifen, 5-fluorouracil, paclitaxel, and doxorubicin were measured using the tetrazolium salt-based proliferation assay (WST assay; Wako Chemicals, Osaka, Japan) according to the manufacturer's instructions. Briefly, $4 \times 10^3$ cells were cultured in 96-well plates in 100 μL of growth medium and incubated for 24 h. Then, 100 μL of medium with a graded concentration of tamoxifen, fluorouracil, paclitaxel, or doxorubicin was added into each well and cultured for 96 h to determine the $IC_{50}$ for the tamoxifen-resistant cells. Then, 10 μL of WST-8 solution was added to each well, and the plates were incubated at 37˚C for another 3 h. The absorbance was measured at 450 nm and 640 nm using SoftMax Pro (Molecular Devices, Tokyo, Japan), and the cell viability was determined. Each experiment was independently performed and repeated at least three times.

## Western blotting

Proteins were isolated from cells as previously described and used for western blot analyses (10 μg/lane) [12, 13]. The membrane was probed with the following antibodies: anti-ERα antibody (1:200; #sc-7207, Santa Cruz Biotechnology, Heidelberg, CA, USA), anti-progesterone receptor (PgR) antibody (1:1000; #sc-810, Santa Cruz Biotechnology), and anti-HER2 antibody (1:1000; #2165S, Cell Signaling Technology, Danvers, MA, USA). β-actin (1:5000; #A5441, Sigma-Aldrich) was used as a loading control. Each experiment was repeated independently at least three times, and one representative blot of each experiment is presented in the figures. Protein levels corresponding to each band were quantified based on band intensity using the ChemiDoc XRS and Quantity One software (Bio-Rad Laboratories, Tokyo, Japan).

## Total RNA extraction and quantitative RT-PCR

Total RNA was extracted using an RNeasy Mini kit (Qiagen, Alameda, CA, USA) according to the manufacturer's instructions. TaqMan® Gene Expression Assays for thymidylate synthetase (*TYMS*) (cat. # Hs00426586_m1), thymidine phosphorylase (*TYMP*) (Hs01034319_g1), dihydropyrimidine dehydrogenase gene (*DPYD*) (Hs00559279_m1), and β-actin (Hs99999903_m1) were purchased from Applied Biosystems, and mRNA levels were quantified in triplicate using the Applied Biosystems 7300 Real-Time PCR system.

## Immunohistochemistry

Sections (3-μm) of paraffin-embedded tumor samples were used for immunohistochemistry. For immunohistochemical analysis, slides were heated for antigen retrieval in 10 mmol/L sodium citrate (pH 6.0). Sections were subsequently exposed to specific antibodies for ERα (Ventana Medical Systems, Tucson, AZ, USA) or dihydropyrimidine dehydrogenase (DPD) (#ab134922; Abcam, Cambridge, UK). Sections were then incubated with Histofine® Simple Stain MAX-PO (MULTI) (Nichirei Biosciences Inc., Tokyo, Japan). Staining was revealed using diaminobenzidine (Nichirei Biosciences Inc), and the slides were counterstained with aqueous hematoxylin.

## Luciferase reporter assays

The 3.0-kb DNA fragment (nt −2918 to +83) including the 5′ region and the noncoding exon 1 of the *DPYD* gene was subcloned into the pGL3-Basic plasmid (Promega, Madison, WI, USA), which encodes firefly luciferase as a reporter (pGL3-DPYDPro3.0), as previously described [14]. To determine whether the *DPYD* gene was post-transcriptionally regulated, the approximately 1.3-kb DNA fragment, including *DPYD* 3′UTR (+3186 bp to +4525 bp downstream of the ATG codon) was subcloned into pGL3-Basic according to a previously reported method [15] and designated as pGL3-DPYD3′UTR. Cells were seeded in six-well plates ($5 \times 10^5$ cells/well) and incubated for 24 h. The pGL3-DPYDPro3.0 (0.2 μg/well) or pGL3-DPYD3′UTR (0.2 μg/well) and Renilla luciferase vector (pRL-SV40; 1 ng/well; Addgene, Watertown, MA, USA) were transiently transfected with TransIT-LT1 transfection reagent (Mirus Bio, Madison, WI, USA) following the manufacturer's protocol. An empty vector (pGL3-Basic) was included as a control in all experiments. Cells were harvested 48 h after transfection in $1 \times$ PLB buffer (Promega), and luciferase activity was measured. All luciferase measurements were normalized to the Renilla readings from the same sample. The experiments were performed in triplicate.

## 5-azacytidine treatment

A total of $5 \times 10^4$ cells were cultured in six-well plates with 5 μM of 5-azacytidine or DMSO (control). After incubation for 96 h, $4 \times 10^3$ cells were cultured in 96-well plates with graded concentrations of fluorouracil for another 96 h. The growth inhibitory effect of 5-fluorouracil was quantitated using a WST assay. For analysis of *DPYD* mRNA expression, cells were immediately frozen as pellets. Total RNA was extracted, and RT-PCR for *DPYD* was performed. β-actin was used as an internal control.

## Experimental mouse model for capecitabine

The Institutional Animal Care and Use Committee of Shinshu University reviewed and approved all the animal experimental procedures in this study (Approval number: 240076), which were conducted according to the recommendations of the United States Public Health

Service Policy on Humane Care and Use of Laboratory Animals (Office of Laboratory Animal Welfare, NIH, Department of Health and Human Services, Bethesda, MD). Six-week-old BALB/c-nu nude female mice weighing 15–18 g were purchased (Charles River Laboratories Japan, Inc., Yokohama, Japan) and were maintained under pathogen-free conditions. Water and food were supplied *ad libitum*. Animals were observed for tumor growth, activity, feeding, and pain according to the guidelines of the Harvard Medical Area Standing Committee on Animals. Capecitabine (Chugai Pharma, Tokyo, Japan) was administered to the mice. Pellets of 17β-estradiol (Sigma-Aldrich) were transplanted into the dorsal region of the mice 5 d before transplantation of MCF7 or MCF7/T cells. Then, $8 \times 10^6$ of MCF7 or MCF7/T cells were injected subcutaneously into another side of the dorsal region. To test the effect of capecitabine, tumor-bearing mice were divided randomly into six groups (n = 3 or 4 per group), when tumor volume was approximately 100–200 mm$^3$. The maximum tolerated dose (MTD) of capecitabine in mice was determined based on data from the manufacturer, which was 539 mg/kg [16]. Each group of mice was administered distilled water only (AQ), 1/2 MTD (269 mg/kg) of capecitabine, or 2/3 MTD (359 mg/kg) of capecitabine orally using an orogastric probe once a day for 5 d, followed by a 2-d washout as one course. Four courses of treatment were performed. Mouse weight was determined every 2 or 3 d. Tumor diameters were measured with a slide caliper every 2 or 3 d, and tumor volume was calculated using the following formula: volume = the major length (mm) × minor length (mm) × minor length (mm)/2. Relative tumor volume (%) was calculated using the following formula: tumor volume at the measuring day/tumor volume at day 1 × 100. Adverse events were judged by body weight (BW) change, which was calculated using the following formula: BW change (%) = [(BW of day n– BW of the classified day)/BW of classified day] × 100.

## Statistical analysis

Data were tested for significance by performing unpaired Student's *t*-tests or one-way ANOVA with Tukey's multiple comparisons; a *p*-value < 0.05 was considered statistically significant as calculated using StatFlex ver. 6 (Artech Co., Ltd., Osaka, Japan).

## Results

### Tamoxifen-resistant sublines in MCF7, T47D and BT474 cells

Three ER-positive breast cancer cell lines, wt-MCF7, wt-T47D, and wt-BT474 were used in the study, and TAM-resistant sublines were designated as MCF7/T, T47D/T, and BT474/T, respectively. The relative tamoxifen resistance of each TAM-resistant subline relative to their corresponding wild-type cell line was determined using a WST assay (Fig 1A, Table 1). The IC$_{50}$ of tamoxifen for wt-MCF7 and TAM-resistant MCF7/T were 4.0 ± 0.7 μM, 10.8 ± 1.1 μM, respectively. The IC$_{50}$ for the wt-T47D and T47D/T was 4.3 ± 1.0 μM and 8.1 ± 1.1 μM, respectively. The IC$_{50}$ for the wt-BT474 and BT474/T was 7.1 ± 1.1 μM and 14.2 ± 4.0 μM, respectively. Thus, MCF7/T, T47D/T, and BT474/T exhibited over 2.7-fold, 1.9-fold, and 2.0-fold higher tamoxifen resistance than their corresponding wild-type cells. We tested the growth inhibitory effects of TAM in wild-type and TAM-resistant sublines of these three cell lines by cell proliferation assay. We observed that each TAM-resistant subline grew in the presence of concentrations of tamoxifen that restricted their corresponding wild-type (S1 Fig).

The expression of estrogen receptor-α (ERα), progesterone receptor (PgR), and HER2 in each wild-type cell line and TAM-resistant subline was evaluated by western blotting (Fig 1B). Expression of ERα was detected in wt-T47D, wt-MCF7, and wt-BT474 cells, although ERα expression in wt-BT474 was lower than that in the other cell lines. In TAM-resistant sublines, the expression of ERα was increased in T47D/T cells. However, the expression of ERα was

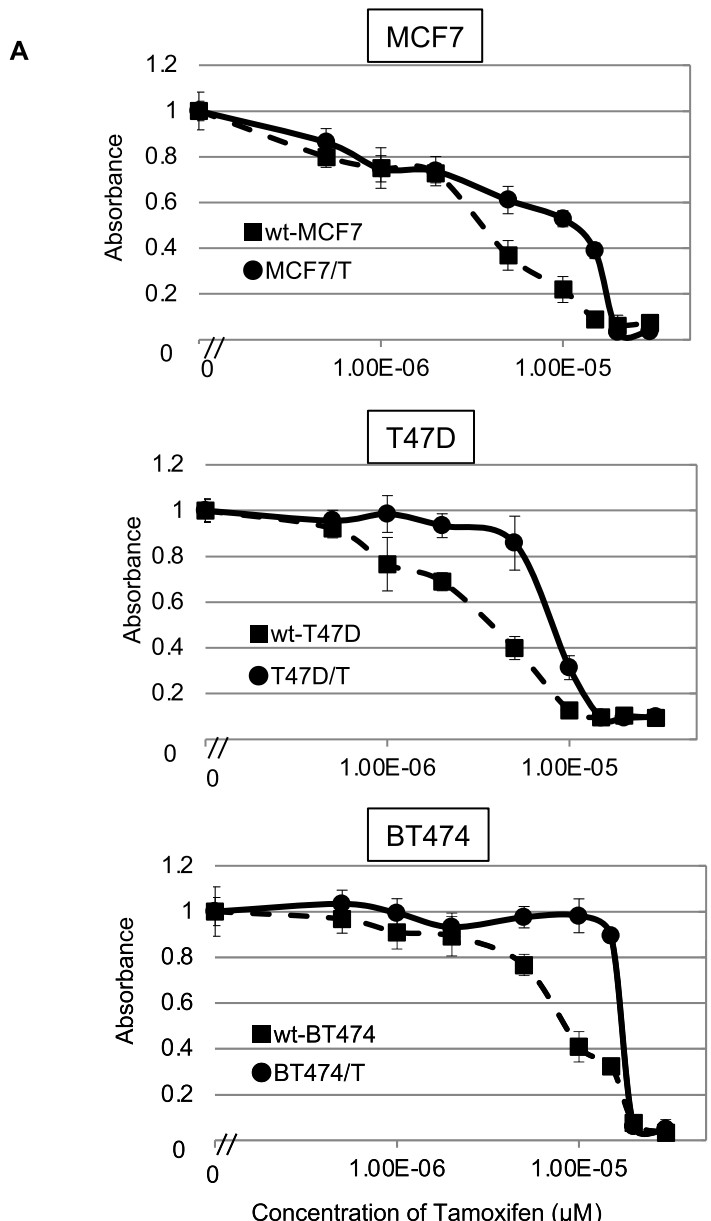

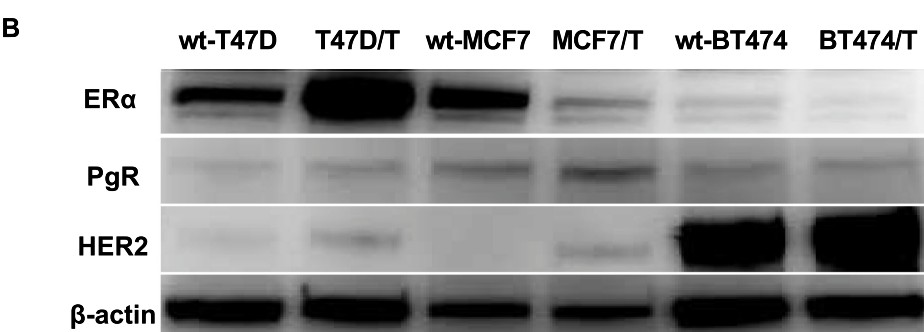

**Fig 1. Growth inhibitory effects of tamoxifen and expression of ERα, PgR and HER2 in ER-positive breast cancer cell lines and their tamoxifen-resistant sublines.** The growth inhibitory effects of TAM in wt-MCF7, MCF7/TAM, wt-T47D, T47D/T, BT474, and BT474/T was evaluated by WST assays (A). Closed circles (●) indicate wild-type cells, whereas closed squares (■) indicate TAM-resistant sublines. The error bars represent the standard errors of the values obtained from triplicate experiments. (B) ERα, PgR, and HER2 protein levels were analyzed by western blotting. β-actin was used as a loading control. Each experiment was independently performed and repeated at least three times, and one representative result is provided in the figures.

decreased in both MCF7/T and BT474/T cells. HER2 expression was increased in all TAM-resistant sublines, although its expression in BT474/T cells, in which the *erbB2* gene was amplified, was remarkably higher than that in T47D/T and MCF7/T cells. With regard to PgR, a slight increase was observed in T47D/T and MCF7/T cells when the cells acquired resistance to TAM.

## Sensitivity to cytotoxic chemotherapeutic agents in wild-type MCF7, T47D, BT474 and their tamoxifen-resistant sublines

To evaluate whether the sensitivity to cytotoxic chemotherapeutic agents was altered when the cells acquired resistance to TAM, we tested whether the sensitivity to 5-fluorouracil, paclitaxel, and doxorubicin was altered in the TAM-resistant sublines (Fig 2). TAM-resistant MCF7 (MCF7/T) cells showed a 15-fold higher increase in sensitivity to 5-fluorouracil (IC$_{50}$, 40 μM) compared with the wt-MCF7 (IC$_{50}$, 600 μM), and a slight increase in sensitivity to doxorubicin was observed in MCF7/T cells compared with the wt-MCF7 cells (IC$_{50}$: 700 μM vs. 150 μM) (Fig 2A). With regard to paclitaxel, no difference in sensitivity was observed between MCF7 and MCF7/T cells. However, no difference in sensitivities to 5-fluorouracil, paclitaxel, and doxorubicin were observed between the wild-type cells and TAM-resistant T47D and BT474 cells (Fig 2B and 2C, and Table 2).

Regarding MCF7 cells, we established several TAM-resistant sublines and tested 5-fluorouracil sensitivity for other clones using WST assays. We found that another representative clone, MCF7/T-2, demonstrated an increased sensitivity to 5-fluorouracil, equivalent to MCF7/T (S2A Fig).

## Comparison of apoptosis induced by cytotoxic chemotherapeutic agents in wild-type and tamoxifen-resistant MCF7 cells

In the present study, a remarkable increase in sensitivity to 5-fluorouracil was observed in the TAM-resistant MCF7 (MCF7/T) subline. To analyze whether the increased sensitivity to 5-fluorouracil in MCF7/T cells was attributable to the increase in apoptosis, the apoptosis

**Table 1. IC$_{50}$ values for tamoxifen for wild-type and tamoxifen-resistant sublines.**

|          | IC$_{50}$ (μM)* | RR ratio** | *p* value |
|----------|-----------------|------------|-----------|
| **MCF7**   | 4.0 ± 0.7  | -   |      |
| **MCF7/T** | 10.8 ± 1.1 | 2.7 | 0.01 |
| **T47D**   | 4.3 ± 1.0  | -   |      |
| **T47D/T** | 8.1 ± 1.1  | 1.9 | 0.01 |
| **BT474**  | 7.1 ± 1.1  | -   |      |
| **BT474/T**| 14.2 ± 4.0 | 2.0 | 0.04 |

*IC$_{50}$: half-maximal inhibitory concentration. mean ± standard deviation

**Relative resistance ratio = IC$_{50}$ of tamoxifen-resistant cells/IC$_{50}$ of wild-type cells.

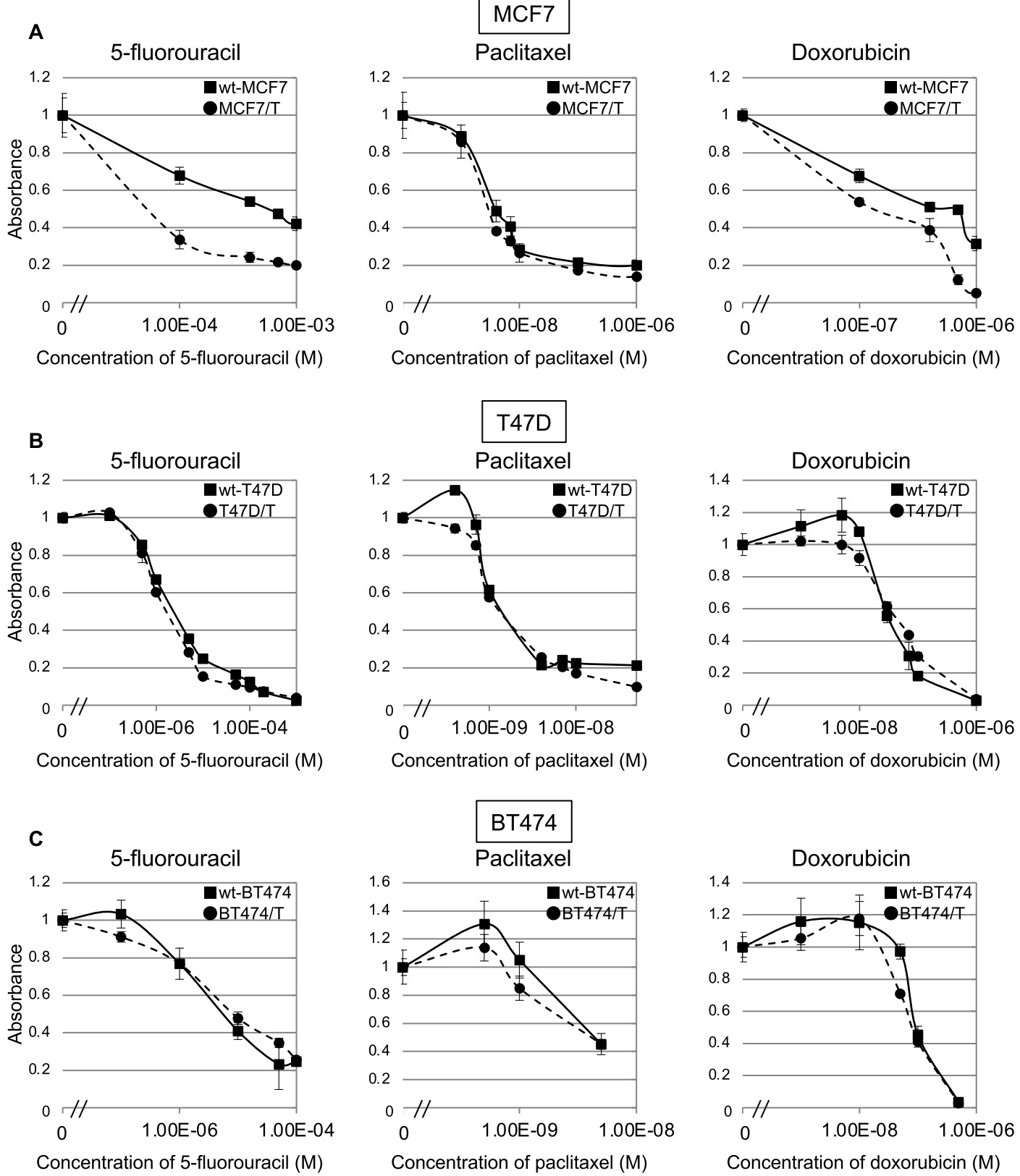

**Fig 2. Sensitivity to chemotherapeutic agents in ER-positive breast cancer cell lines and their tamoxifen-resistant sublines.** Sensitivity to 5-fluorouracil (a), paclitaxel (b), and doxorubicin (c) in wild-type (wt) and TAM-resistant MCF7 (A), T47D (B), and BT474 (C) cells were determined using the WST assay. Black lines with closed squares (■) indicate wild-type cells (wt-MCF7, wt-T47D, and wt-BT474), dotted lines with closed circles (●) indicate TAM-resistant sublines (MCF7/T, T47D/T, and BT474/T). Error bars represent standard deviations of the values obtained from triplicate experiments. Each experiment was independently performed and repeated at least three times, and one representative result is provided in the figures.

induced by three cytotoxic chemotherapeutic agents was compared quantitatively in wt-MCF7 and MCF7/T cells by detecting DNA fragmentation (Fig 3). After 48 h of culture with 5-fluorouracil at concentrations from 50 μM to 2 mM, DNA fragmentation was not detected in the wt-MCF7 cells. In contrast, a dose-dependent increase in the levels of DNA fragmentation was observed in MCF7/T cells after 48 h of treatment with 5-fluorouracil. With regard to paclitaxel, no difference in DNA fragmentation induced by the drug was observed between the wt-MCF7 and MCF7/T cells at concentrations from 5 nM to 1 μM, which was consistent with the drug sensitivity profile obtained by the WST assay. As for doxorubicin, significantly higher levels of DNA fragmentation were observed in MCF7/T cells than the wt-MCF7 cells, which was consistent with the results observed in the WST assay. These results indicate that apoptosis induced by 5-fluorouracil and doxorubicin was increased in TAM-resistant MCF7 cells.

## Thymidine synthase, thymidine phosphorylase, and dihydropyrimidine dehydrogenase expression in wild-type MCF7, T47D, BT474 and their tamoxifen-resistant cell lines

The sensitivity to 5-fluorouracil was remarkably increased in TAM-resistant MCF7 cells. As a remarkable increase in sensitivity to 5-fluorouracil was observed in TAM-resistant MCF7 cells, we focused on the analysis of the expression of the molecules involved in the metabolic pathway of 5-fluorouracil. 5-fluorouracil is converted to fluorodeoxyuridine (FdUrd) by thymidine phosphorylase, and is then phosphorylated by thymidine kinase to fluorodeoxyuridine monophosphate (FdUMP). Inhibition of thymidylate synthase by FdUMP is one of the principal mechanisms of 5-fluorouracil's action [17]. However, 5-fluorouracil is enzymatically inactivated by DPD to form dihydrofluorouracil (DHFU). Subsequently, DHFU is metabolized to α-fluoro-ureidopropionic acid (FUPA), then 2-fluoro-β-alanine (FBAL). releasing ammonia and carbon dioxide [17]. Based on this background, we evaluated whether the mRNA expression of thymidylate synthase, thymidine phosphorylase, and *DPYD* was altered in the TAM-resistant breast cancer sublines (Fig 4).

**Table 2. IC$_{50}$ values for 5-fluorouracil, paclitaxel, and doxorubicin for wild-type and tamoxifen-resistant sublines.**

| Cell line | Chemotherapeutic agents | | | | | |
|---|---|---|---|---|---|---|
| | 5-fluorouracil | | Paclitaxel | | Doxorubicin | |
| | IC$_{50}$ (μM)* | RR ratio** | IC$_{50}$ (μM)* | RR ratio** | IC$_{50}$ (μM)* | RR ratio** |
| **wt-MCF7** | 566.7 ±164.9 | - | 2.7 ±0.3 | - | 303.3 ±68.4 | - |
| **MCF7/T** | 37.0 ±2.9 | 0.07 | 2.4 ±0.3 | 0.88 | 89.3 ±15.1 | 0.29 |
| **wt-T47D** | 1.6 ±0.2 | - | 0.9 ±0.1 | - | 30.0 ±2.1 | - |
| **T47D/T** | 1.3 ±0.2 | 0.81 | 1.0 ±0.1 | 1.1 | 38.8 ±2.4 | 1.29 |
| **wt-BT474** | 4.3 ±1.0 | - | 3.7 ±0.2 | - | 85.3 ±10.1 | - |
| **BT474/T** | 5.4 ±1.0 | 1.26 | 3.3 ±0.4 | 0.89 | 73.3 ±4.7 | 0.88 |

IC$_{50}$: half-maximal inhibitory concentration

* mean ± standard deviation

**Relative resistance ratio = IC$_{50}$ of anticancer drug-resistant cells/IC$_{50}$ of wild-type cells

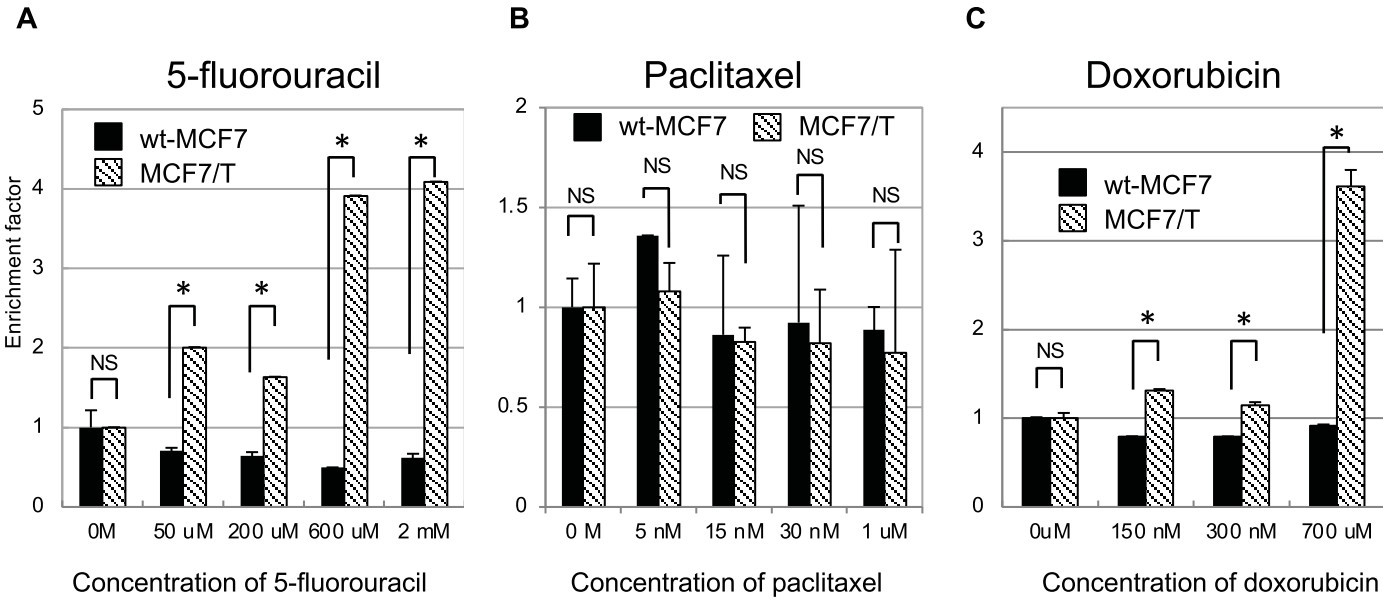

**Fig 3. Apoptosis induced by chemotherapeutic agents in ER-positive breast cancer cell lines and their tamoxifen-resistant sublines.** The effect of 5-fluorouracil (A), paclitaxel (B), and doxorubicin (C) on apoptosis in wild-type and TAM-resistant MCF7, T47D, and BT474 cells was examined. The Cell Death Detection ELISA plus kit was used to quantify apoptosis in the presence of 5-fluorouracil (50 μM–2 mM), paclitaxel (5 nM–1 μM), or doxorubicin (150 nM–700 μM) for 48 h. The error bars represent the standard deviations of the values obtained; each experiment was performed in duplicate. The experiments were repeated independently at least three times, and one representative result is provided in the figures. NS, not significant; *$p < 0.05$, unpaired Student's $t$-tests.

The mRNA expression of thymidine synthase was significantly decreased in all TAM-resistant sublines (Fig 4A). As for thymidine phosphorylase mRNA, MCF7/T and T47D/T cells showed a significantly higher level of expression than the wild-type cells (Fig 4B). *DPYD* mRNA expression was significantly higher in the wt-MCF7 cells than in the other two wild-type cells. However, *DPYD* mRNA expression was drastically decreased in MCF7/T cells (Fig 4C). A remarkable decrease of *DPYD* mRNA expression was observed in another TAM-resistant MCF7 subline, MCF7/T-2, as well (S2B Fig).

To confirm whether thymidine synthase or DPD were involved in the susceptibility of MCF-7 cells to 5-fluorouracil, we tested whether the knockdown of either enzyme would alter 5-fluorouracil sensitivity in wt-MCF7 cells (S3 Fig). Inhibition of *DPYD* mRNA expression by siRNA sensitized the wt-MCF7 cells to 5-fluorouracil. In addition, we quantitated the intracellular concentrations of 5-fluorouracil metabolites, FdUrd and FBAL, in wt-MCF and MCF7/T cells by matrix-assisted laser desorption/ionization time-of-flight mass spectrometry (MALDI-TOF MS) (S4 Fig). We found that the amount of 5-fluorouracil's active metabolite, FdUrd, was higher in MCF7-T cells compared to wt-MCF7 cells, while that of FBAL was lower.

These data suggest that the decrease in the target enzyme thymidine synthase, together with a drastic decrease in its catabolic enzyme, DPD, may enhance the efficacy of 5-fluorouracil in MCF7/T cells.

## Promoter activity of *DPYD* gene in wild-type MCF7 and tamoxifen-resistant MCF7 cells

As a significant decrease in *DPYD* mRNA expression was observed in the TAM-resistant MCF7 (MCF7/T) cells, we examined the promoter activity of the *DPYD* gene in wt-MCF7 and MCF7/T cells using luciferase reporter analysis. The luciferase reporter activity driven by the

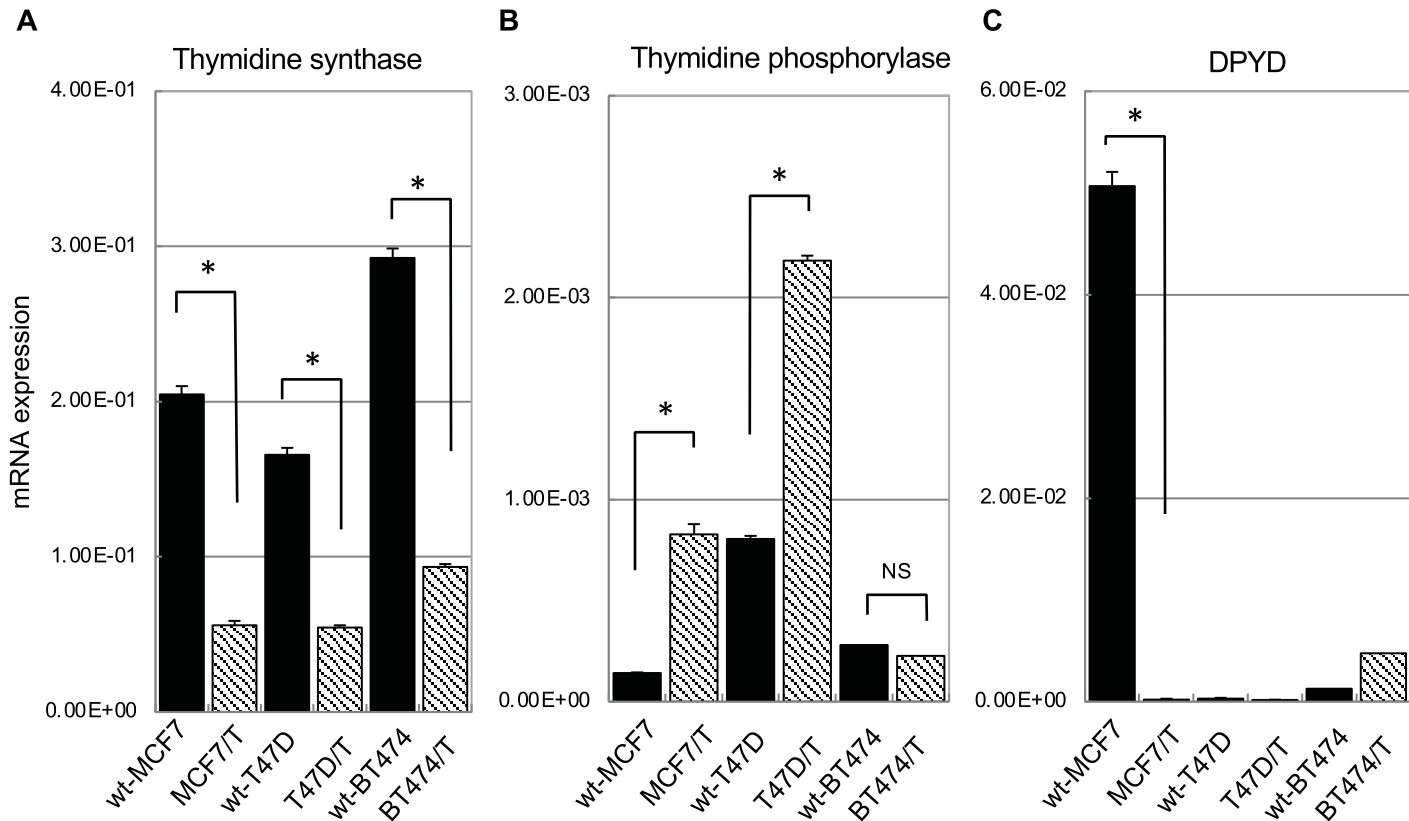

**Fig 4. The expressions of thymidine synthase, thymidine phosphorylase, and dihydropyrimidine dehydrogenase in ER-positive breast cancer cell lines and their tamoxifen-resistant sublines.** The mRNA expression of thymidine synthase (A), thymidine phosphorylase (B), and *DPYD* (C) were quantified by real-time RT-PCR in ER-positive wild-type cell lines (MCF7, T47D, and BT474) and their TAM-resistant sublines (MCF7/T, T47D/T, and BT474/T). β-actin was used as an internal control. The error bars in each graph represent the standard deviations of the values obtained in the experiments performed in triplicate. The experiments were repeated independently at least three times, and one representative result is provided in the figures. NS, not significant; $^*p < 0.05$, by unpaired Student's *t*-tests.

5′ region of *DPYD* was significantly increased in MCF7/T cells compared with the wt-MCF7 cells (Fig 5).

## Demethylation by 5-azacytidine treatment and expression of *DPYD mRNA*

There was a discrepancy between exogenous promoter activity and *DPYD* mRNA expression levels in TAM-resistant MCF7 cells. One of the likely mechanisms involved in the discrepancy is a genetic mutation in the promoter region; however, no mutation was detected in the sequence of the 5′ region of *DPYD*. Hence, we hypothesized that some factors might interfere with the post-transcriptional regulation of *DPYD* mRNA and tested whether an epigenetic alteration in the promoter region, that is, aberrant methylation, was involved in the transcriptional regulation of *DPYD*. To examine whether promoter methylation participates in the transcriptional repression of *DPYD* in wt-MCF7 and MCF7/T cells, we performed reverse analysis of DNA methylation with a demethylating agent, 5-azacytidine treatment in these cells. The expression of *DPYD* mRNA was restored with 5 μM 5-azacytidine treatment after 96 h (Fig 6A). Treatment with 5 μM 5-azacytidine for 96 h increased *DPYD* mRNA expression up to 1.2 and 8.6 times in wt-MCF7 and MCF7/T cells, respectively. These data indicated the possibility that abnormal hypermethylation was involved in the decrease of *DPYD* mRNA expression observed in MCF7/T cells, which showed a higher *DPYD* promoter activity, and may partly

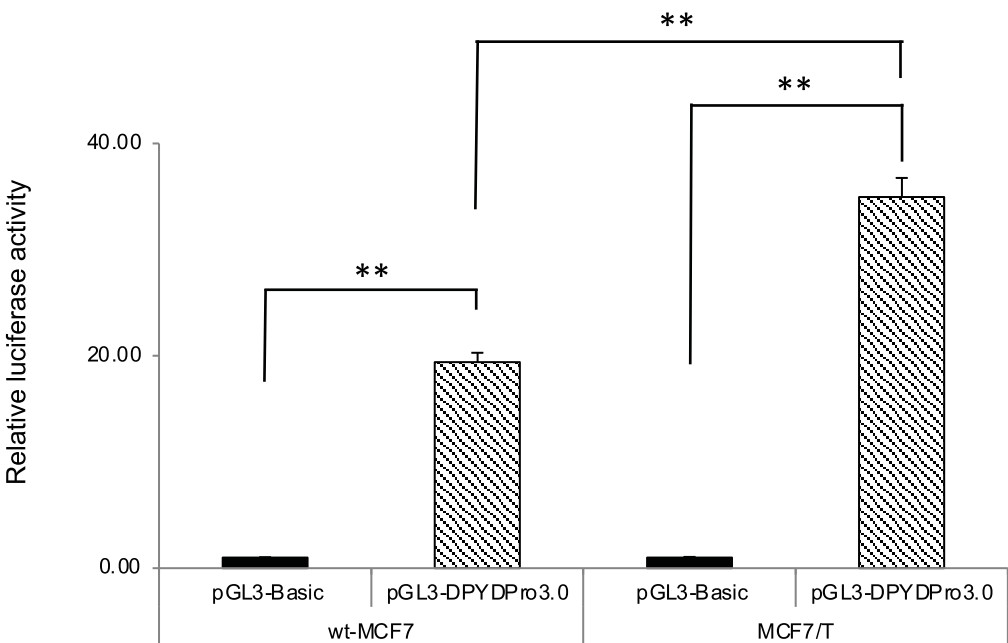

**Fig 5. *DPYD* promoter activity in wild-type MCF7 and tamoxifen-resistant MCF7 cells.** Exogenous promoter activity of *DPYD* was measured by transient transfection assay of the 5′ region of *DPYD* with the luciferase reporter gene. Relative luciferase activity normalized to pGL3-Basic in each cell line is expressed. The experiments were repeated independently at least three times, and one representative result is provided in the figures. NS: not significant, $^{**}p < 0.01$, using unpaired Student's *t*-tests.

explain the discrepancy between *DPYD* mRNA expression level and its promoter activity observed in MCF7/T cells.

Next, we tested whether treatment with 5-azacytidine alters the sensitivity to 5-fluorouracil of MCF7/T cells by WST assay. As demonstrated in Fig 6B, when the MCF7/T cells were treated with 5 μM of 5-azacytidine, a decrease of 5-fluorouracil sensitivity was observed (IC50 for wt-MCF7; 15 μM, MCF7/T; 21 μM). These data indicated the possibility that hypermethylation-mediated modulation of *DPYD* mRNA expression may partly be involved in altering the sensitivity of MCF7/T cells to 5-fluorouracil.

### *DPYD* 3′-UTR activity in wild-type MCF7 and tamoxifen-resistant MCF7 cells

To investigate whether post-transcriptional regulation by miRNA was involved in the decreased *DPYD* gene expression observed in TAM-resistant MCF7 (MCF7/T) cells, we tested *DPYD* 3′-UTR reporter activity by 3′-UTR luciferase assay (Fig 7). The *DPYD* 3′-UTR luciferase activity in MCF7/T cells was significantly lower than that in wt-MCF7 cells ($p < 0.05$). These results suggested that post-transcriptional regulation by miRNAs may also be involved in the decreased *DPYD* mRNA expression observed in MCF7/T cells.

### Effect of capecitabine on wild-type MCF7 and tamoxifen-resistant MCF7 cells in tumor xenograft model

Next, we investigated whether TAM-resistant MCF7 (MCF7/T) cells showed higher sensitivity to 5-fluorouracil in the mouse xenograft model. We chose capecitabine for the treatment of

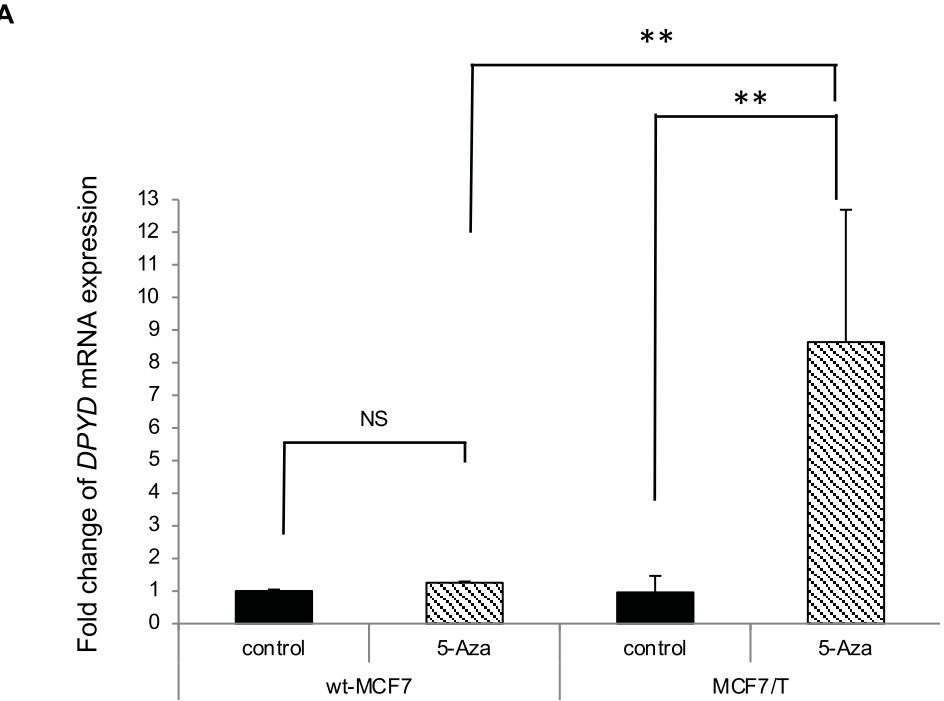

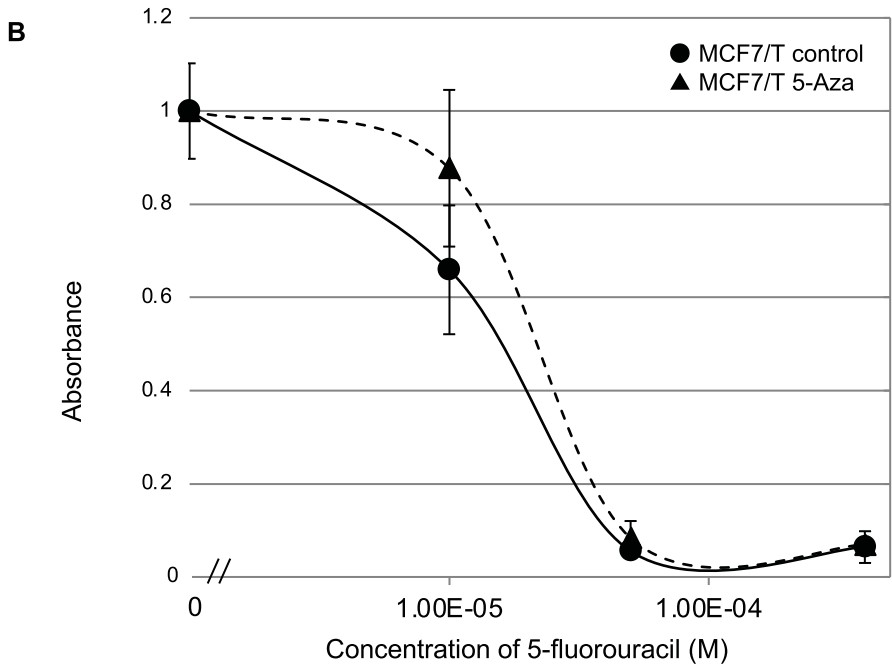

**Fig 6. Effects of a demethylating agent on *DPYD* mRNA expression and sensitivity to tamoxifen in wild-type MCF7 and tamoxifen-resistant MCF7 cells.** Alteration of *DPYD* mRNA expression and sensitivity to 5-fluorouracil exerted by a demethylating agent, 5-azacytidine, was tested in wild-type and TAM-resistant MCF7 (MCF7/T) cells. (A)

*DPYD* mRNA expression in wt-MCF7 and MCF7/T cells treated with 5 μM 5-azacytidine for 96 h was analyzed by real-time RT-PCR. Relative expression levels were calculated as ratios of the expression in the treated cells to those in untreated cells. The error bars represent the standard deviations of the values obtained in the experiments performed in triplicate. The experiments were repeated independently at least three times, and one representative result is provided in the figures. NS not significant, $^{**}p < 0.01$ by one-way ANOVA with Tukey's multiple comparisons. (B) Effects of 5-azacytidine treatment on sensitivity to 5-fluorouracil was tested in MCF7/T cells using WST assay. The black line with closed circles (●) indicates control, and the dotted line with closed triangles (▲) indicates cells treated with 5 μM of 5-azacytidine. Error bars represent standard deviations of the values obtained from triplicate experiments. Each experiment was independently performed and repeated at least three times, and one representative result is provided in the figures.

tumors in the xenograft. As capecitabine, an orally administered prodrug of 5-fluorouracil, is selectively activated by tumor cells to its cytotoxic moiety, 5-fluorouracil and capecitabine monotherapy has been used globally to treat recurrent breast cancer [16–18]. The use of capecitabine was considered a closer model for clinical breast cancer (Fig 8).

Both MCF7 and MCF7/T cells were inoculated subcutaneously at the dorsal region of the mice, and the anti-tumor effect of 5-fluorouracil was tested in the tumor xenograft model by oral administration of capecitabine. Before starting the administration of capecitabine, the expression of ERα and DPD was evaluated by immunohistochemistry, and immunohistochemical analysis demonstrated that the expression of DPD in the MCF7/T tumor was remarkably lower than that in the MCF7 tumor, indicating that the tumor created by subcutaneous inoculation of two cell lines maintained the characteristic observed *in vitro* (Fig 8A).

In the mice bearing wt-MCF7 tumors, capecitabine inhibited tumor growth in a dose-dependent manner at a dose of 1/2 MTD and 2/3 MTD; however, the tumor continued to

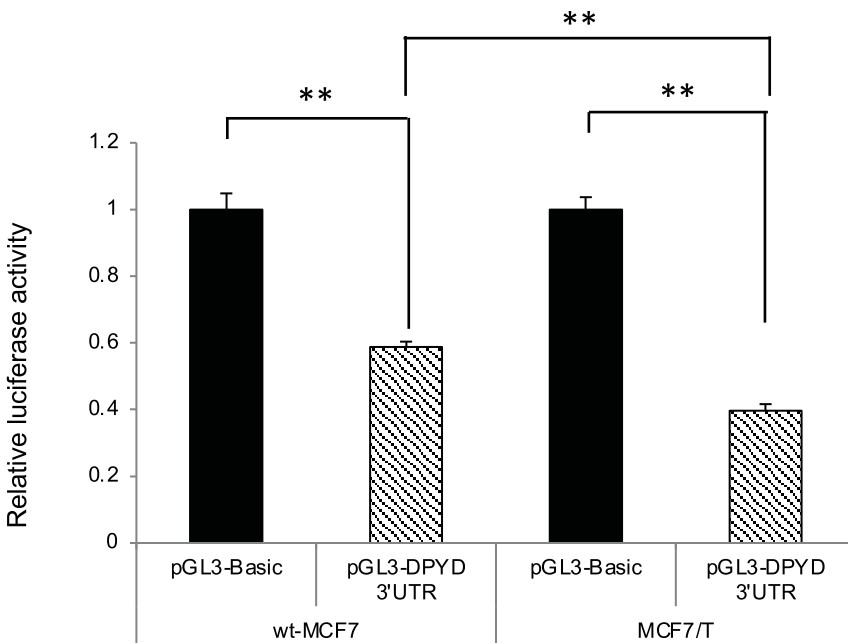

**Fig 7. *DPYD* 3′-UTR activity in wild-type MCF7 and tamoxifen-resistant MCF7 cells.** Wild-type (wt)-MCF7 and TAM-resistant MCF7 (MCF7/T) cells were transfected with the pGL3-DPYD3′ UTR, and luciferase activity was measured as described in the Materials and Methods. The data shown were normalized with the internal control. The error bars represent the standard deviations of the values obtained in the experiments performed in triplicate. The experiments were repeated independently at least three times, and one representative result is provided in the figures. NS, not significant; $^{**}p < 0.01$ by one-way ANOVA with Tukey's multiple comparisons.

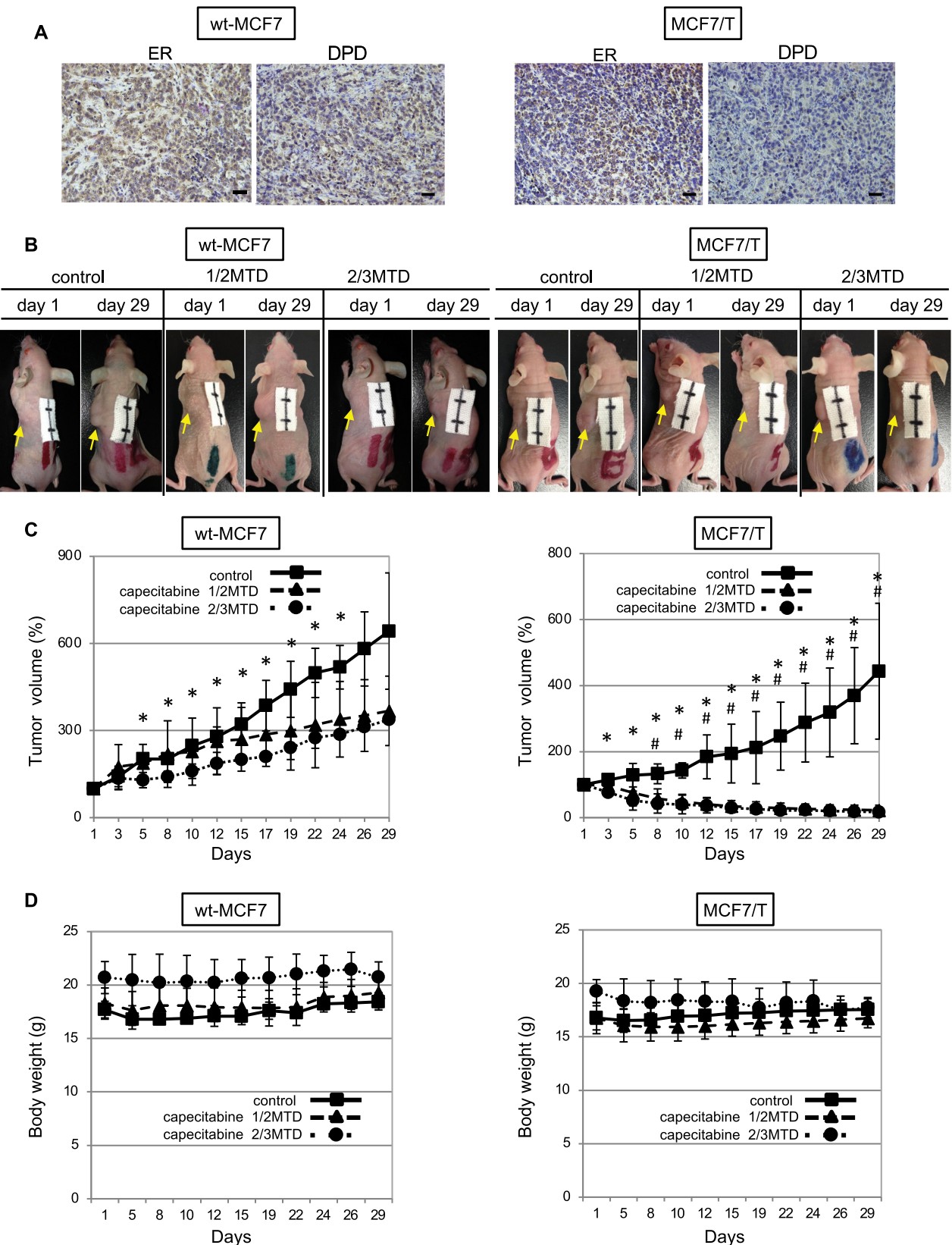

**Fig 8. Anti-tumor effects of capecitabine in mouse xenograft models.** The anti-tumor effect of capecitabine was tested in the wt-MCF7 and TAM-resistant MCF7 (MCF7/T) tumor xenograft model. Distilled water (control), 1/2 maximum tolerated dose (MTD) of capecitabine, or 2/3 MTD of capecitabine were orally administered with an orogastric probe once a day for five days, and then they were given a two-day washout as one course. Four courses of treatment were performed. (A) Representative photographs of immunohistochemistry (×200) for ERα and DPD in tumors obtained from control groups on day 29. Scale bars = 100 μm. (B) Representative photographs of mice bearing wt-MCF7-tumor (*left*) and MCF7T-tumor (*right*) in each treatment group on days 1 and 29. Each scale bar represents 1 cm. (C) The mean tumor volumes were plotted from day 1 to day 29, with measurements taken every two or three days (*left*; wt-MCF7 tumors, *right*; MCF7/T tumors). Closed squares (■) indicate control, closed triangles (▲) indicate 1/2 MTD of capecitabine, and closed circles (●) indicate 2/3 MTD groups. $*p < 0.01$ (control group vs. 2/3 MTD group), $\# p < 0.01$ (control group vs. 1/2 MTD group) using unpaired Student's *t*-tests. (D) The mean body weights were plotted from day 1 to day 29. Closed squares (■) indicate control, closed triangles (▲) indicate 1/2 MTD of capecitabine, and closed circles (●) indicate 2/3 MTD groups.

grow in the presence of these doses of capecitabine (Fig 8B and 8C). In contrast, a reduction in tumor volume was observed from the early phase of treatment with 1/2 MTD of capecitabine in the mice bearing MCF7/T tumors, and almost no tumors were detected 4 weeks after the initiation of 1/2 or 2/3 MTD of capecitabine (Fig 8B and 8C). No body weight changes were observed throughout the treatment of any of the groups. Thus, a significant increase in sensitivity to capecitabine was observed in the MCF7/T cells introduced into the *in vivo* mouse xenograft model.

## Discussion

There has been no "gold standard therapy" established for metastatic breast cancer, and the therapeutic strategy for each patient is usually decided by considering both patient and disease characteristics as well as previous treatments [19]. Because endocrine therapy is often performed for a long period in patients with ER-positive breast cancer, there is a possibility that previous endocrine therapy affects expression or function of the molecules related to the sensitivity to chemotherapeutic agents. In the present study, we established TAM-resistant sublines in three ER-positive breast cancer cell lines (MCF7, T47D, and BT474) and demonstrated that TAM-resistant MCF7 (MCF7/T) cells showed a higher sensitivity to 5-fluorouracil than wt-MCF7 cells, and an alteration of molecules associated with the metabolic pathway of 5-fluorouracil was induced in TAM-resistant cells. The biological interaction between TAM and chemotherapeutic agents has long been investigated, and conflicting observations on this interaction in terms of anti-tumor activity have been reported [20–23]. For example, Kurebayashi reported that short-term exposure to 4-OH-TAM or estradiol depletion reduced thymidine synthase expression, while a combination of both 4-OH-TAM with 5-fluorouracil and estradiol depletion with 5-fluorouracil enhanced the growth inhibitory effect in ER-positive KPL-1 cells [21, 22]. However, there have been few findings on the effects of long-term administration of endocrine therapeutic agents on the sensitivity of ER-positive breast cancer to subsequent administration of chemotherapeutic agents. Thus, to the best of our knowledge, this is the first study that demonstrates the possibility of modification of sensitivity to 5-fluorouracil by primary resistance to TAM in ER-positive breast cancer cells. In addition, our results indicate that the changes induced by long-term TAM administration differ between cell lines, suggesting that more personalized treatment strategies are required in clinical recurrent cancers.

There are three main active metabolites of 5-fluorouracil: fluorodeoxyuridine monophosphate (FdUMP), fluorodeoxyuridine triphosphate (FdUTP), and fluorouridine triphosphate (FUTP). Thymidine phosphorylase converts 5-fluorouracil to fluorodeoxyuridine (FdUrd), which is then phosphorylated to FdUMP by thymidine kinase. Subsequently, inhibition of thymidylate synthase by FdUMP inhibits the activity of thymidylate synthase, which leads to interference of DNA synthesis. However, DPD is the rate-limiting enzyme in the 5-fluorouracil catabolism [17, 24], and DPD mediates conversion of 5-fluorouracil to dihydrofluorouracil (DHFU) in normal and tumor cells. To increase the bioavailability and efficacy of

5-fluorouracil, DPD-inhibitory fluoropyrimidines have been developed and clinically applied for over 30 years [25, 26]. Thus, thymidine synthase, thymidine phosphorylase, and DPD are known to be involved in the sensitivity of cancer cells to 5-fluorouracil. In the present study, the expression of thymidine synthase was decreased in all TAM-resistant sublines established from three different ER-positive cell lines, and the expression of thymidine phosphorylase was increased in TAM-resistant MCF7 and T47D cells. However, the only TAM-resistant subline established from MCF7 became significantly susceptible to 5-fluorouracil. Grem *et al.* reported that cell lines with lower *DPYD* mRNA expression tended to be more susceptible to 5-fluorouracil. Meanwhile, neither thymidine synthase expression nor thymidine kinase activity correlated with the growth inhibitory effect of 5-fluorouracil in the analyses examining the association between the growth inhibitory effect of 5-fluorouracil and the expression of thymidine synthase, thymidine kinase, and DPYD in 63 cancer cell lines including 11 breast cancer cell lines [27]. Our findings, together with the report by Grem *et al.*, suggest that a significantly decreased DPD expression might confer susceptibility to 5-fluorouracil in TAM-resistant MCF7.

In the present study, we found that the expression of *DPYD* mRNA was repressed by both methylation of the *DPYD* promoter region and post-transcriptional regulation by miRNA, at least in part. These findings are consistent with the results of previous studies on the association between the regulatory mechanisms of the *DPYD* gene and sensitivity to 5-fluorouracil in cancer cells [14, 15].

Accumulating evidence has indicated that various epigenetic mechanisms are involved in TAM resistance in luminal-type breast cancer cells [28–30]. However, few reports have analyzed how epigenetic modulation induced in TAM-resistant luminal type breast cancer cells affects the susceptibility to chemotherapeutic agents. Hence, a novel finding in our study is that epigenetic alterations induced in breast cancer cells over the course of development of resistance to tamoxifen could modify the susceptibility to subsequent cytotoxic agents.

Over the past 20 years, oral fluorouracil derivatives have been developed. These oral derivatives enable the fluorouracil concentration to be increased in the tumor by *in vivo* enzymatic conversion while avoiding gastrointestinal toxicity. Capecitabine and S-1 are both oral fluorouracil derivatives that have been widely used in the treatment of breast cancer. Either of the drugs has been shown to have therapeutic effects on metastatic breast cancer as a single agent in a randomized control trial [11, 31–33]. Orally administered fluorouracil derivatives are generally more convenient than intravenous cytotoxic agents, and they allow patients to avoid hair loss, which is usually the most distressing adverse effect of chemotherapy [34]. Thus, oral fluorouracil derivatives have some advantages and are expected to remain important treatment options. In a randomized control trial comparing the efficacy of oral capecitabine versus a classical regimen cyclophosphamide, methotrexate, and fluorouracil (CMF) as first-line chemotherapy for women with advanced breast cancer who were unsuited to more intensive regimens, capecitabine improved overall survival compared with CMF [35]. Moreover, the results of this RCT demonstrated that the hazard ratio for the comparison of overall survival between capecitabine and CMF was significantly lower in ER-positive patients than in negative patients. In the present study, we demonstrated the alteration of the expression of enzymes related to 5-fluorouracil metabolism by long-term administration of TAM. Our findings suggest the possibility that the greater therapeutic effect of capecitabine in ER-positive patients observed in this RCT reflected the modification of the enzymatic activity related to 5-fluorouracil metabolism by prior endocrine therapies, as demonstrated in the TAM-resistant cells in the present study.

Here, increased 5-fluorouracil susceptibility after acquisition of TAM resistance was observed in MCF7 alone among three ER-positive breast cancer cell lines. Hence, it is not clear

whether the decreased expression of *DPYD* mRNA is attributed to the characteristics of the breast cancer cell line itself or otherwise to the dose or duration of treatment with tamoxifen. However, decreased expression of thymidine synthase was observed in all three cell lines, suggesting that long-term TAM administration may alter gene expression involved in the sensitivity to chemotherapeutic drugs in various breast cancer cells.

However, the mRNA expression of genes involved in the sensitivity to 5-fluorouracil differed between MCF7 and the other two cell lines, representing the individual diversity observed in clinical breast cancer. These findings suggest the indispensability of evaluating biomarkers to develop an appropriate treatment strategy for patients with recurrent breast cancer and resistance to therapeutic agents.

## Conclusions

In the present study, we demonstrated that the preceding long-term tamoxifen administration could alter the sensitivity to subsequent chemotherapeutic agents such as 5-fluorouracil in some ER-positive breast cancer cell lines. Our results suggest that 5-fluorouracil and its derivatives may act as critical drugs in some TAM-resistant ER-positive breast cancers.

A limitation of this study was that we were not able to elucidate whether reduced *DYPD* mRNA expression was causally linked to TAM administration. Further research is required to elucidate the precise mechanism of how TAM alters the *DPYD* mRNA expression in ER-positive breast cancer cells.

## Supporting information

**S1 Fig. Growth curves of wild-type and tamoxifen-resistant sublines in the presence of various concentrations of tamoxifen.** The growth inhibitory effects of TAM in wt-MCF7, MCF7/TAM, wt-T47D, T47D/T, BT474, and BT474/T was evaluated by cell proliferation assay. (A) The growth of wild-type and TAM-resistant MCF7, T47D, and BT474 cells treated with tamoxifen was measured by direct cell count. The relative proliferation rate was plotted by comparing the number of cells at each time point with the number at 0 h. The error bars represent the standard deviations of the values obtained from triplicate experiments. (PDF)

**S2 Fig. Characteristics of two tamoxifen-resistant sublines established from MCF cells.** We had established several TAM-resistant sublines for MCF7 cells, and we tested 5-fluorouracil sensitivity by WST assay (A), and *DPYD* mRNA expression by real-time RT-PCR (B) in a representative clone, MCF7/T-2. MCF7-T2 demonstrated an increased sensitivity to 5-fluorouracil equivalent to MCF7/T, and showed a decreased expression of *DPYD* mRNA compared to wt-MCF7 cells. (PDF)

**S3 Fig. Alteration of 5-fluorouracil sensitivity by thymidine synthase or *DPYD* knockdown in wild-type MCF7 cells.** To evaluate whether thymidine synthase (TS) or dihydropyrimidine dehydrogenase (DPYD) were involved in sensitivity to 5-fluorouracil, we tested whether the knockdown of either enzyme would alter 5-fluorouracil sensitivity in wt-MCF7 cells. Inhibition of *TS* and *DPYD* mRNA expression was confirmed by real-time RT-PCR (A, B). The sensitivity to 5-fluorouracil was tested by WST assay (C). siRNA targeting of *DPYD* sensitized the wt-MCF7 cells to 5-fluorouracil, while siRNA targeting of *TS* did not alter the sensitivity to 5-fluorouracil. (PDF)

**S4 Fig. Quantitation of 5-fluorouracil metabolites in wild-type and tamoxifen-resistant MCF7 cells.** The intracellular concentrations of 5-fluorouracil metabolites, fluorodeoxyuridine (FdUrd, *left panel*) and 2-fluoro-β-alanine (FBAL, *right panel*) were quantitated by matrix-assisted laser desorption/ionization time-of-flight mass spectrometry (MALDI-TOF MS) as described in the **S1 File**. The amount of 5-fluorouracil active metabolite, FdUrd and FBAL were higher and lower in MCF7-T cells compared with those in wt-MCF7 cells, respectively. The experiment was done in duplicate.
(PDF)

**S1 File. Supplementary materials and methods.**
(DOCX)

**S2 File. Uncropped images of western blots.**
(PDF)

## Acknowledgments

We would like to thank Editage (www.editage.jp) for English language editing.

## Author Contributions

**Conceptualization:** Takayuki Watanabe, Tomohiro Shibata, Ken-ichi Ito.

**Data curation:** Takayuki Watanabe, Takaaki Oba, Ken-ichi Ito.

**Formal analysis:** Takayuki Watanabe.

**Funding acquisition:** Ken-ichi Ito.

**Investigation:** Takayuki Watanabe, Takaaki Oba, Shinobu Kamijo, Ken-ichi Ito.

**Methodology:** Takayuki Watanabe, Takaaki Oba, Keiji Tanimoto, Tomohiro Shibata, Ken-ichi Ito.

**Project administration:** Takayuki Watanabe, Ken-ichi Ito.

**Software:** Takaaki Oba.

**Supervision:** Ken-ichi Ito.

**Writing – original draft:** Takayuki Watanabe, Takaaki Oba, Ken-ichi Ito.

**Writing – review & editing:** Tomohiro Shibata, Ken-ichi Ito.

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
