## [Decision Letter · Decision Letter 0]

26 Feb 2021

PONE-D-21-03437

Tamoxifen resistance alters sensitivity to 5-fluorouracil in a subset of estrogen receptor-positive breast cancer

PLOS ONE

Dear Dr. Ito,

Thank you for submitting your manuscript to PLOS ONE. After careful consideration, we feel that it has merit but does not fully meet PLOS ONE’s publication criteria as it currently stands. Therefore, we invite you to submit a revised version of the manuscript that addresses the points raised during the review process.

Specifically, the reviewers suggested to examine 5-FU sensitivity in multiple breast cancer cell lines and to perform knockdown of essential enzymes.

We look forward to receiving your revised manuscript.

Kind regards,

Wei Xu

Academic Editor

PLOS ONE

Journal Requirements:

Reviewers' comments:

Reviewer's Responses to Questions

**Comments to the Author**

1. Is the manuscript technically sound, and do the data support the conclusions?

Reviewer #1: Yes

Reviewer #2: Yes

2. Has the statistical analysis been performed appropriately and rigorously? 

Reviewer #1: Yes

Reviewer #2: Yes

3. Have the authors made all data underlying the findings in their manuscript fully available?

Reviewer #1: Yes

Reviewer #2: Yes

4. Is the manuscript presented in an intelligible fashion and written in standard English?

Reviewer #1: Yes

Reviewer #2: Yes

5. Review Comments to the Author

Reviewer #1: This is a very interesting study where the authors examine the sensitivity to 5-FU after ER positive cells acquire tamoxifen resistance.

Below are my comments to further complete the study and clarify the mechanisms of drug sensitivity to 5-FU:

1. One representative tamoxifen resistant clone was used in this study. However, the growth rate of parental and tamoxifen resistant sub-line is different for MCF7, T47D and BT-474. MCF7-T cells growth rate was almost doubled compared with parental, while T47D-T and BT-474-T decreased compared with parental lines. It seems that the sensitivity to 5-FU or other chemotherapy drugs correlated with the growth rate of the cells (parental and tamoxifen resistant subclone). Since there is only one cell line that has preferential sensitivity to 5-FU after acquiring tamoxifen resistance, the authors need to carefully validate observations seen in MCF7 using multiple clones (and also T47D and BT-474), to clarify whether this is due to the growth rate of the subclones or not.

2. In order to obtain the conclusion that “decrease in the target enzyme thymidine synthase, together with a drastic decrease in its catabolic enzyme, dihydropyrimidine dehydrogenase, may enhance the efficacy of 5-fluorouracil in MCF7/T cells”, the authors need to directly knockdown thymidine synthase and DPYD in parental MCF7, T47D and BT-474 cells and test for the sensitivity to 5-FU. Otherwise, this observation is merely a correlation but did not indicate the causative effect for 5-FU sensitivity.

3. If 5-aza restores the level of DPYD, did it lead to resistance to 5-FU treatment in MCF7-T cells?

Reviewer #2: About one third of ER positive breast cancer patients showed tamoxifen resistance after a long-term treatment. However, few studies have investigated the response to subsequent chemotherapy in ER positive breast cancer which showed resistance to endocrine therapy. The objective of this study is to focus on whether preceding endocrine therapy could alter the sensitivity to chemotherapeutic agents in ER-positive breast cancers and find biomarkers for personalized treatment that acquired resistance endocrine therapy. The manuscript by Takayuki Watanabe et al. selected three tamoxifen resistant cell lines (T47D/T, MCF7/T and BT474/T), and then tested cells sensitivity to chemotherapeutic agents compared with wild type (wt) cells. The results showed MCF7/T cells became more sensitive to 5-fluorouracil than wt-MCF7 cells. This is because the expression of dihydropyrimidine dehydrogenase (DPYD) was decreased in MCF/T cells compared with wt-MCF7 cells. DPD is an important enzyme to degrade 5-fulorouracil to inactive metabolites. Mechanismly, they found the expression of DYPD was repressed by methylation of its promoter and post-transcriptional regulation by miRNA in MCF7/T cells. They also performed in vivo xenograft experiments to confirm capecitabine could significantly reduce tumor volume in MCF7/T compared with MCF7 cells. These new findings of this paper demonstrated that 5-fulorouracil and its derivatives may act as critical drugs in some TAM-resistant ER positive breast cancers. Overall the experimental designs are rigor with mouse models and the results support their conclusions. A few minor comments are listed below:

1. BT474 cell line is always used as a de novo tamoxifen resistant model, so it’s not appropriate for the author to select tamoxifen resistant cell line (BT474/T).

2. Fig. 1 looks crowded and confused, drawing a curve to show the response to different concentrations in each WT and resistant cell line (just like figure 2) will be more nice and clearly.

3. For the result of Fig. 4, the author should show a little more background information about these enzymes, thymidine synthase, thymidine phosphorylase and dihydropyrimidine dehydrogenase, explain the correlation between the expression of these enzymes and the cells sensitive to 5-fluorouracil. What’s more, the author should also test the metabolites of 5-fulorouracil in wt-MCF7 and MCF7/T cells.

4. In Fig 8, the staining of ER and DPD by IHC is unclear, the author should repeat and show a clear picture.

6. PLOS authors have the option to publish the peer review history of their article (what does this mean?). If published, this will include your full peer review and any attached files.

Reviewer #1: No

Reviewer #2: No

---

## [Author Response · Author response to Decision Letter 0]

10 May 2021

Point-by-point responses to the reviewers’ comments

We thank the reviewers for their helpful and insightful comments. We revised our manuscript in accordance with these comments, and we believe our manuscript has greatly improved, thanks to the reviewers’ feedback. We hope that the newly submitted manuscript is now suitable for publication in PLOS ONE.

Reviewer #1: 

This is a very interesting study where the authors examine the sensitivity to 5-FU after ER positive cells acquire tamoxifen resistance. Below are my comments to further complete the study and clarify the mechanisms of drug sensitivity to 5-FU:

1. One representative tamoxifen resistant clone was used in this study. However, the growth rate of parental and tamoxifen resistant sub-line is different for MCF7, T47D and BT-474. MCF7-T cells growth rate was almost doubled compared with parental, while T47D-T and BT-474-T decreased compared with parental lines. It seems that the sensitivity to 5-FU or other chemotherapy drugs correlated with the growth rate of the cells (parental and tamoxifen resistant subclone). Since there is only one cell line that has preferential sensitivity to 5-FU after acquiring tamoxifen resistance, the authors need to carefully validate observations seen in MCF7 using multiple clones (and also T47D and BT-474), to clarify whether this is due to the growth rate of the subclones or not.

Response:

We appreciate the reviewer’s insightful comments. To evaluate tamoxifen (TAM) resistance more objectively, we have re-examined TAM sensitivity for wild-type and TAM-resistant sublines with a WST assay and demonstrated the results in Figure 1A in the revised manuscript. In addition, we have also presented TAM IC50s for wild-type and TAM-resistant sublines in Table 1. From these results, we consider the difference of TAM sensitivity between wild types and our established resistant sublines to not be solely due to differences in the growth rate. We have described the results mentioned above in the first paragraph on page 12 of the revised manuscript.

In the cell proliferation assay presented in Figure 1A of the originally submitted version, the initial cell number was different for each cell line (ex. wt-MCF7, 14 × 104; MCF7/T, 45 × 104). Therefore, a large difference was observed in the number cells grown after 120 h. However, when calculating the growth rate, there was no remarkable difference between wild-type and TAM-resistant sublines. The growth curve presented as Figure 1A in the originally submitted version is attached as Supplementary figure 1, with the vertical axis corrected to the growth rate for reference.

Regarding MCF7, we had established several TAM-resistant sublines. We have included the data obtained using one subline, MCF7/T2, as Supplementary figure 2. As shown in Supplementary figure 2A, MCF7-T2 showed an increased sensitivity to 5-fluorouracil, equivalent to MCF7/T, while decreased expression of DPYD mRNA was observed in MCF7/T2, as demonstrated in Supplementary figure 2B. We consider these results to show that a decrease in DPYD mRNA expression may be one of the changes induced in the development of TAM resistance in MCF7. We have described the results mentioned above in the second paragraph on page 14 and in the second paragraph on page 18 of the revised manuscript.

2. In order to obtain the conclusion that “decrease in the target enzyme thymidine synthase, together with a drastic decrease in its catabolic enzyme, dihydropyrimidine dehydrogenase, may enhance the efficacy of 5-fluorouracil in MCF7/T cells”, the authors need to directly knockdown thymidine synthase and DPYD in parental MCF7, T47D and BT-474 cells and test for the sensitivity to 5-FU. Otherwise, this observation is merely a correlation but did not indicate the causative effect for 5-FU sensitivity.

Response:

We appreciate the reviewer’s insightful comments. Per the reviewer’s suggestion, we tested whether knockdown of thymidine synthase (TS) or dihydropyrimidine dehydrogenase (DPYD) would alter the sensitivity to 5-fluorouracil in wt-MCF7 cells. Although the reviewer mentioned that we should test the knockdown of TS and DPYD in wt-T47D and at-BT474 cells as well, we performed the additional experiments in wt-MCF7 cells alone because the baseline expression level of DPYD mRNA was not high enough in wt-T47D and at-BT474 cells to inhibit its expression by siRNA. As demonstrated in Supplementary figure 3, inhibition of TS and DPYD expression by siRNA was confirmed at the mRNA level for wt-MCF7 cells (A, B). siRNA targeting of DPYD sensitized the wt-MCF7 cells to 5-fluorouracil, while siRNA targeting of TS did not alter the sensitivity to 5-fluorouracil (C). We have described the results mentioned above in the third paragraph on page 18 of the revised manuscript.

3. If 5-aza restores the level of DPYD, did it lead to resistance to 5-FU treatment in MCF7-T cells?

Response:

We appreciate the reviewer’s insightful comments. Per the reviewer’s suggestion, we tested whether treatment with 5-azacytidine alter the sensitivity to 5-fluorouracil of TAM-resistant MCF7 (MCF7/T) cells. We have included the results of the WST assay as Figure 6B. As demonstrated in Figure 6A, when the MCF7/T cells were treated with 5 μM of 5-azacytidine, a decrease of 5-fluorouracil sensitivity was observed in parallel with an increased DPYD mRNA expression. These data indicate the possibility that hypermethylation-mediated modulation of DPYD mRNA expression may partly be responsible for altering the sensitivity of MCF/T cells to 5-fluorouracil. We have described the results mentioned above in the second paragraph on page 21. 

Reviewer #2: 

About one third of ER positive breast cancer patients showed tamoxifen resistance after a long-term treatment. However, few studies have investigated the response to subsequent chemotherapy in ER positive breast cancer which showed resistance to endocrine therapy. The objective of this study is to focus on whether preceding endocrine therapy could alter the sensitivity to chemotherapeutic agents in ER-positive breast cancers and find biomarkers for personalized treatment that acquired resistance endocrine therapy. The manuscript by Takayuki Watanabe et al. selected three tamoxifen resistant cell lines (T47D/T, MCF7/T and BT474/T), and then tested cells sensitivity to chemotherapeutic agents compared with wild type (wt) cells. The results showed MCF7/T cells became more sensitive to 5-fluorouracil than wt-MCF7 cells. This is because the expression of dihydropyrimidine dehydrogenase (DPYD) was decreased in MCF/T cells compared with wt-MCF7 cells. DPD is an important enzyme to degrade 5-fulorouracil to inactive metabolites. Mechanismly, they found the expression of DYPD was repressed by methylation of its promoter and post-transcriptional regulation by miRNA in MCF7/T cells. They also performed in vivo xenograft experiments to confirm capecitabine could significantly reduce tumor volume in MCF7/T compared with MCF7 cells. These new findings of this paper demonstrated that 5-fulorouracil and its derivatives may act as critical drugs in some TAM-resistant ER positive breast cancers. Overall the experimental designs are rigor with mouse models and the results support their conclusions. A few minor comments are listed below:

1. BT474 cell line is always used as a de novo tamoxifen resistant model, so it’s not appropriate for the author to select tamoxifen resistant cell line (BT474/T).

Response:

We appreciate the reviewer’s comments. As the reviewer stated, the IC50 for TAM of BT474 cells was higher than those of wt-MCF7 and wt-T47D cells (Table 1); however, we successfully established TAM-resistant sublines for BT474 cells, wherein the IC50 was 2.0 times higher than that of the wild type. Hence, we would like to keep the results obtained for BT474 cells in the revised manuscript. 

2. Fig. 1 looks crowded and confused, drawing a curve to show the response to different concentrations in each WT and resistant cell line (just like figure 2) will be more nice and clearly.

Response:

We agree with the reviewer’s point. We have re-examined TAM sensitivity for wild-type and TAM-resistant sublines with the WST assay and have demonstrated the results as Figure 1A in the revised manuscript. The growth curve presented as Figure 1A in the originally submitted version is attached as Supplementary figure 1, with the vertical axis corrected to the growth rate for reference. We have described the results in the first paragraph on page 12 of the revised manuscript.

3. For the result of Fig. 4, the author should show a little more background information about these enzymes, thymidine synthase, thymidine phosphorylase and dihydropyrimidine dehydrogenase, explain the correlation between the expression of these enzymes and the cells sensitive to 5-fluorouracil. What’s more, the author should also test the metabolites of 5-fulorouracil in wt-MCF7 and MCF7/T cells.

Response:

We appreciate the reviewer’s insightful comments. 5-fluorouracil is converted to fluorodeoxyuridine (FdUrd) by thymidine phosphorylase, and is then phosphorylated by thymidine kinase to fluorodeoxyuridine monophosphate (FdUMP). Inhibition of thymidylate synthase by FdUMP is one of the principal mechanisms of 5-fluorouracil’s action. However, 5-fluorouracil is enzymatically inactivated by dihydropyrimidine dehydrogenase to form dihydrofluorouracil (DHFU) (DPD). Subsequently, DHFU is metabolized to α-fluoro-ureidopropionic acid (FUPA), then 2-fluoro-β-alanine (FBAL), releasing ammonia and carbon dioxide. Based on this background, we analyzed the mRNA expression of thymidylate synthase, thymidine phosphorylase, and DPYD in wild-type and TAM-resistant sublines.

In accordance with the helpful suggestion by the reviewer, we have provided concise background information about the metabolism of 5-fluorouracil, as mentioned in the second paragraph on page 17 of the Results. We have edited the description of interpretation in the second paragraph on page 27 in the Discussion as well.

Moreover, we analyzed the metabolites of 5-fluorouracil in wt-MCF7 and MCF7/T cells, following the reviewer’s suggestion. We quantitated the two 5-fluorouracil metabolites (FdUrd and FBAL) by matrix-assisted laser desorption/ionization time-of-flight mass spectrometry (MALDI-TOF MS). We observed that the amount of 5-fluorouracil’s active metabolite, FdUrd, was higher in MCF7-T cells compared to wt-MCF7 cells, while that of FBAL was lower. These data demonstrate high sensitivity to 5-fluorouracil observed in MCF7/T cells. We have included these data as Supplementary figure 4 in the revised manuscript. We have described the results mentioned above in the third paragraph on page 18 of the revised manuscript.

4. In Fig 8, the staining of ER and DPD by IHC is unclear, the author should repeat and show a clear picture.

Response:

We appreciate the reviewer’s helpful comment. We have replaced the pictures in Figure 8A with higher resolution images.

---

## [Decision Letter · Decision Letter 1]

24 May 2021

Tamoxifen resistance alters sensitivity to 5-fluorouracil in a subset of estrogen receptor-positive breast cancer

PONE-D-21-03437R1

Dear Dr. Ito,

We’re pleased to inform you that your manuscript has been judged scientifically suitable for publication and will be formally accepted for publication once it meets all outstanding technical requirements.

Kind regards,

Wei Xu

Academic Editor

PLOS ONE

Reviewers' comments:

Reviewer's Responses to Questions

**Comments to the Author**

1. If the authors have adequately addressed your comments raised in a previous round of review and you feel that this manuscript is now acceptable for publication, you may indicate that here to bypass the “Comments to the Author” section, enter your conflict of interest statement in the “Confidential to Editor” section, and submit your "Accept" recommendation.

Reviewer #1: All comments have been addressed

Reviewer #2: All comments have been addressed

2. Is the manuscript technically sound, and do the data support the conclusions?

Reviewer #1: Yes

Reviewer #2: Yes

3. Has the statistical analysis been performed appropriately and rigorously? 

Reviewer #1: Yes

Reviewer #2: Yes

4. Have the authors made all data underlying the findings in their manuscript fully available?

Reviewer #1: Yes

Reviewer #2: Yes

5. Is the manuscript presented in an intelligible fashion and written in standard English?

Reviewer #1: Yes

Reviewer #2: Yes

6. Review Comments to the Author

Reviewer #1: (No Response)

Reviewer #2: All comments had been addressed, the manuscript had been revised according to the comments. The manuscript could be accepted.

7. PLOS authors have the option to publish the peer review history of their article (what does this mean?). If published, this will include your full peer review and any attached files.

Reviewer #1: No

Reviewer #2: No

---

## [Editor Report · Acceptance letter]

31 May 2021

PONE-D-21-03437R1 

Tamoxifen resistance alters sensitivity to 5-fluorouracil in a subset of estrogen receptor-positive breast cancer 

Dear Dr. Ito:

I'm pleased to inform you that your manuscript has been deemed suitable for publication in PLOS ONE. Congratulations! Your manuscript is now with our production department. 

Kind regards, 

on behalf of

Dr. Wei Xu 

Academic Editor

PLOS ONE